# Comparative Analysis of Gene and MicroRNA Expression in Subcutaneous Adipose Tissue in Metabolically Healthy and Unhealthy Obesity

**DOI:** 10.3390/ijms26178212

**Published:** 2025-08-24

**Authors:** Natalia O. Markina, Georgy A. Matveev, Ksenia A. Zasypkina, Natalia V. Khromova, Alina Yu. Babenko, Evgeny V. Shlyakhto

**Affiliations:** Science Department of Metabolic Deviations and Personalized Prevention, Almazov National Medical Research Centre, St. Petersburg 197341, Russiafmrc@almazovcentre.ru (E.V.S.)

**Keywords:** metabolically healthy obesity, subcutaneous adipose tissue, microRNA, adipose tissue hypoxia, insulin resistance

## Abstract

Metabolically healthy (MHO) and unhealthy obesity (MUO) exhibit distinct molecular genetic mechanisms underlying metabolic disorders. Studying gene and microRNA expression in subcutaneous adipose tissue (SAT) may reveal key pathogenetic differences between these phenotypes. We compared the expression of genes (ADIPOQ, HIF1A, CCL2) and microRNAs (miR-142-3p, miR-155, miR-378) in SAT between MHO and MUO patients and assessed their association with metabolic parameters. The study included 39 obese patients (19 MHO, 20 MUO) and 10 healthy controls. SAT biopsies were analyzed using real-time PCR. Correlations with clinical and metabolic markers were evaluated. Obese patients showed decreased ADIPOQ (*p* = 0.039) and miR-142 (*p* = 0.008) expression and increased CCL2 (*p* = 0.004), miR-155 (*p* = 0.017), and miR-378 (*p* = 0.04) expression compared to the controls. MUO patients exhibited higher HIF1A expression (*p* = 0.03) and strong correlations between CCL2 and dyslipidemia (total cholesterol, triglycerides)/dysglycemia (fasting glucose) (r = 0.45, *p* = 0.03; r = 0.52, *p* = 0.01; r = 0.63, *p* = 0.001, respectively). miR-142 negatively correlated with fibrosis markers, while miR-378 was linked to insulin resistance. The differential gene and microRNA expression highlights the role of inflammation, hypoxia, and fibrosis in MUO pathogenesis. miR-142-3p, miR-155, and miR-378 may serve as potential biomarkers for metabolic risk stratification and therapeutic targets.

## 1. Introduction

The timing and intensity of the development of metabolic disorders in obese patients vary widely and do not always directly depend on the duration and severity of obesity. Understanding the processes occurring in adipose tissue at the molecular genetic level can shed light on the causes of these differences. To date, it has been established that the expansion of adipose tissue upon excess energy intake can occur in two main ways, i.e., physiologically, involving hyperplasia of adipocytes in the subcutaneous fat depot, and pathologically, characterized by hypertrophy of adipocytes and the accumulation of excess energy not only and not so much in the subcutaneous depot as in the visceral one.

Currently, many studies are attempting to characterize changes in gene expression in adipose tissue with different types of fat accumulation and also to compare the characteristics of subcutaneous adipose tissue (SAT) and visceral adipose tissue [1,2]. One of the most popular hypotheses explaining the “switch” of fat deposition from subcutaneous to visceral depots is that according to which, with the volume of subcutaneous fat increase, the blood supply deficit and hypoxia increase, which leads to inflammation and fibrosis. The latter limits the plasticity of SAT and restricts its further expansion. This hypothesis suggests that in order to understand the mechanisms of the changes occurring, the comparison of the SAT characteristics in different phenotypes of obesity at the stage of preserved metabolic health and during the development of metabolic disorders is of great interest. We attempted to implement such an approach in our study by examining some molecular genetic characteristics of SAT in patients with metabolically healthy obesity (MHO) and metabolically unhealthy obesity (MUO). We chose the Aguilar-Salinas criteria [3] due to their ease of use, consistency with NCEP ATP III, and high accuracy in the diagnosis of MHO/MHO. These criteria are presented in Table 1.

The analysis included genes involved in the cellular differentiation of adipose tissue (AT) (ADIPOQ), the response to hypoxia (Hif1a), inflammation (Ccl2), and browning (ADIPOQ). Recent studies have demonstrated that obesity changes the profile of a number of miRNAs in AT [4,5], including miRNA-142-3p, miRNA-378, miRNA-155 [6,7]. So, miRNA-378, in a number of studies, is designated as an insulin signaling inhibitor via the inhibition of p110a, which is a glucose and fatty acid metabolism regulator. It has been shown that miRNA-378 is involved in the regulation of insulin sensitivity through counteracting PGC1b and is associated with the development of insulin resistance [8]. It also modulates the expression of proangiogenic and proinflammatory proteins [9]. However, other studies have established a connection between miRNA-378 and the activation of brown adipose tissue during adipogenesis [10], which may determine its participation in maintaining metabolic health. In white adipocytes, overexpression of miRNA-378 promotes an increase in lipid droplets size [11] and hence the depositing ability.

MiRNA-142 has been associated with inflammation activation in older studies, but the related genetic pathway has not been established. Recent studies have demonstrated its involvement in the TGF-β and its receptor inhibition pathways [12], antifibrotic effects [13], the reduction in the formation of type 1 collagen following its overexpression [14], and the inhibition of fat cell differentiation and autophagy [15]. There is evidence that miRNA-142 expression is associated with tumor growth suppression [16]. Our earlier studies found increased expression of this miRNA in SAT in obesity [17].

Recent studies have shown that miRNA-155 mediates obesity-induced inflammation and renal dysfunction [18]. Its overexpression in pancreatic β-cells impairs insulin secretion and enhances β-cell proliferation [19]. Overexpression of miRNA-155 in adipose tissue has been shown to worsen fibrosis in mice [20] and enhance inflammation in adipose tissue [7]. Thus, miRNAs represent a modern tool allowing for better understanding molecular genetic pathways involved in the realization of various effects. In this regard, the assessment of the expression in SAT of the above-mentioned microRNAs regulating the activity of key genes (miR-142-3p, miR-155, miR-378) was included in the analysis. Our study aimed to determine the differences in the expression of these molecular genetic markers in MHO and MUO.

## 2. Results

### 2.1. Participants’ Characteristics

Thirty-nine obese patients and ten healthy controls completed the study. The main characteristics of the examined group are presented in Table 2.

The detailed characteristics of the included obese patients are presented in Table 3.

### 2.2. Differential Gene and miRNA Expression Profiles in Obesity: Associations with Metabolic Dysregulation

We performed an expression analysis of the studied genes and miRNAs in the entire group of obese patients and compared the results with the expression in the healthy control group.

The results are presented in Table 4.

When comparing the expression of genetic markers in obese patients with that in the HCG, there was a significant decrease in the level of adiponectin gene expression by more than twice (*p* = 0.039) and a significant increase in the expression of the hypoxia marker CCL2 by three times (*p* = 0.004) and the inflammation marker miR-155 by two times (*p* = 0.01). No statistically significant differences in the expression of the hypoxia-inducible factor HIF1awere found. The expression of miR-142 was significantly reduced compared to that in the HCG (*p* = 0.008), and the expression of miR-378 was significantly increased (*p* = 0.04).

The analysis did not reveal significant relationships between the expression of ADIPOQ genes and age, BMI, carbohydrate and lipid metabolism indicators, and circulatory inflammatory response biomarkers (CRP) in the general group. HIF1a expression positively correlated with CCL2 expression (r = 0.45, *p* = 0.02) and strongly negatively correlated with miR-142 expression (r = −0.98, *p* = 0.002). The expression of the inflammatory marker CCL2 had a positive correlation with fasting blood glucose level (r = 0.42, *p* = 0.01), TC (r = 0.44, *p* = 0.01), LDL (r = 0.37, *p* = 0.04), TGs (r = 0.53, *p* = 0.03). MiR-155 expression positively correlated with fasting blood glucose level (r = 0.75, *p* = 0.01) and HOMA-B index (r = 0.66, *p* = 0.03) in women. No other relationships with miR-155 expression were revealed. In general, unidirectional correlations were observed between the expression of inflammation and hypoxia markers and metabolic disorders, primarily regarding higher glucose and TG levels.

### 2.3. Distinct Molecular Patterns in MHO and MUO: Inflammation, Hypoxia, and Metabolic Correlations

We then performed a comparative analysis of the expression of the studied molecular genetic markers in the groups with MHO and MUO. The characteristics of the patients in the MHO and MUO groups are presented in Table 5.

There was no significant difference in BMI and the ratio of men and women in the MHO and MUO groups, and the differences in age were on the verge of reliability, but there were regular differences in metabolic parameters (glucose, insulin, TGs, HDL, CRP). The results of the expression assessment of the studied molecular genetic markers are presented in Figure 1a–e.

As follows from the presented data, both in patients with MUO and in patients with MHO, the level of expression of inflammation marker genes (Ccl2 and miR155) was increased compared to that in the control group. The hypoxia-inducible factor gene expression (Hif1a) was increased only in patients with MUO (*p* = 0.03), while in patients with MHO it did not differ from the level of expression in the HCG. In the MUO group, a strong positive relationship between CCL2 expression and metabolic parameters was noted, in contrast to the MHO group (Table 6).

The expression levels of miR-378 were significantly higher in the MUO group (*p* = 0.01) compared to the control group, and the expression of miR-142 was significantly lower compared to the control group both in the MUO group (*p* ≤ 0.01) and in the MHO (*p* = 0.01) group, although the severity of the decrease was significantly higher in MUO patients (*p* = 0.01). In MHO patients, a positive relationship was observed between miR-142 and WHR (r = 0.77, *p* = 0.01) and miR-378 and HDL (r = 0.52, *p* = 0.02). In MUO patients, a positive relationship was observed between miR-142 and HDL (r = 0.55, *p* = 0.04), and a negative relationship between miR-378 and GFR (r = −0.47, *p* = 0.04).

## 3. Discussion

Adipose tissue is an important participant in pathological changes occurring in obesity. The data obtained in our study indicate a connection between increased expression of the inflammation markers CCL2 and miR-155, the hypoxia marker HIF1a, and the appearance of metabolic syndrome components, in particular those of carbohydrate and lipid metabolism disorders (dyslipidemia, dysglycemia). It should be noted that inflammation markers (CCL2 and miR-155) were increased in both MHO and MUO. The level of the circulatory inflammation marker hsCRP was also increased in both groups, although it was significantly higher in the MUO group.

CCL2 (C-C motif ligand 2) is a cytokine belonging to the β-chemokine group, produced in a number of tissues, including adipose tissue, fibroblasts, and mononuclear phagocytes. Moreover, in earlier studies it was noted that an increase in its expression and concentration in adipocytes leads to an enlarged recruitment of macrophages to the adipose tissue, and an increase in its level in the circulation occurs concomitantly with an enhancement in the concentration of tumor necrosis factor alpha (TNF-α) and other proinflammatory cytokines [21]. Experimental studies have demonstrated that CCL2 stimulates the local proliferation of macrophages, forming a proinflammatory microenvironment in visceral adipose tissue [22]. A significant increase in CCL2 expression in adipose tissue in an experimental study significantly reduced the ability of insulin to stimulate glucose transport into cells, which led to hyperglycemia and an increase in LDL and triglyceride levels [23]. These data are a clear premise to our results, which demonstrated that inflammatory changes not only in visceral but also in subcutaneous AT are associated with changes in metabolic parameters. Although we noted only a slight trend towards higher CCL2 expression in SAT in the presence of MUO, strong relationships between the expression of this marker and metabolic parameters (glucose, cholesterol, and TG levels) were noted specifically in the MUO group.

According to Karkeni’s study [7], miR-155 expression is upregulated in adipocytes in the presence of proinflammatory factors (TNF-alpha, IL-6, LPS) in obese individuals and exacerbates chronic inflammation. This miRNA is known to impair adipocyte function by antagonizing PPAR-gamma and C/EBP-beta, which reduces fat cell differentiation and increases insulin resistance. Mouse studies have shown that miR-155 knockout affects insulin sensitivity, which supports the important role of this miRNA in metabolic disorders and provides a direct link between miR-155 levels and the obesity phenotype [24].

MiR-155 expression deficiency is characterized by the preservation of normal insulin sensitivity and a low risk of atherosclerosis. In experimental studies, it was noted that miR-155 overexpression in β-cells disrupted insulin secretion and enhanced β-cell proliferation, while its hyperexpression in adipose tissue aggravated fibrosis. These data are in harmony with our results showing higher miR-155 expression in subcutaneous adipose tissue in obese patients compared to healthy individuals and a relationship between its expression, glucose levels, and beta-cell secretory function, which, however, was noted only in women. This is possibly due to the proposed link between female sex hormones, the expression of their receptors, and the expression of miR-155.

One study on gender differences in microRNA expression and their impact on cardiovascular disease risk [25] describes the mechanisms behind these discrepancies. In this study, the authors state that the expression of non-coding RNAs is sex-dependent as a result of two main factors. First, many gene promoters contain estrogen-responsive elements (EREs, up to 41), through which E2 binding stimulates microRNA expression. Secondly, the X chromosome encodes 118 types of miRNAs, and several of the X-linked miRNAs are known to escape inactivation on the X chromosome, which leads to higher expression levels of these miRNAs in certain cell types. The above may explain our results. Thus, the expression of miRNA-155 may have been higher in women, resulting in stronger associations of its expression with metabolic parameters compared to the male group.

The high expression of HIF1a demonstrated in our study only in MUO suggests that tissue hypoxia may play an important role in the transformation of MHO into MUO. As shown in the study by Seo et al. [26], chronic adipose tissue hypoxia in obesity, exacerbated by mitochondrial dysfunction through increased ANT2 (adenine nucleotide translocase 2) expression in adipocytes, activates HIF1α, which in turn promotes inflammation and metabolic disorders, creating a vicious cycle of insulin resistance. We also noted a consistency in the changes in HIF1α expression and CCL2 expression (r = 0.45, *p* = 0.02) and a strong negative relationship with antifibrogenic miR-142 expression, which confirms the role of hypoxia in the SAT in inflammation development and fibrogenesis induction. Increased HIF1α expression can be considered a signaling marker of the transformation of MHO into MUO.

In our study we found a significant decrease in miR-142 expression in the SAT of obese patients compared to the control group. Meanwhile, the data from individual studies that assessed its level vary. Thus, in studies examining the level of miR-142 in blood, including our earlier study [17], miR-142 level was elevated. In the study presented by Ortega FJ et al. [6], obese patients showed a steep increase in circulating miR-142-3p levels, which correlated with the appearance of the main components of metabolic syndrome, i.e., insulin resistance (HOMA-IR) and systemic inflammation (elevated CRP levels). The few studies that, like ours, assessed miR-142 expression in SAT found it to be decreased in obese patients. However, in the study by Wei et al. [15], miR-142-3p content in serum and the expression of this microRNA in the SAT of obese patients were significantly lower than in healthy volunteers. A possible mechanism was described in the publication by Rakib et al. [27]. The article emphasizes that obesity causes immune cell hyperactivation (M1, Th17 macrophages) as part of a chronic inflammatory process, which naturally leads to a massive release of proinflammatory cytokines (TNF-α, IL-6), which in turn can suppress miR-142 biogenesis through the activation of nuclear factor kappa B (NF-κB) (excess NF-κB disrupts pre-microRNA processing), which can also accelerate miR-142 degradation through RNA polymerase induction. Differences in the rates at which these processes occur may explain the differences in our results. In addition, in a number of studies, miR-142 demonstrated antifibrogenic properties [13,14,28]. This provides miR-142 with an important role in counteracting adipose tissue fibrosis.

MiR-378 has a number of seemingly multidirectional effects on metabolism. It inhibits insulin signaling via glucose and fatty acid metabolism regulation, promoting the development of insulin resistance (IR) [8], and is involved in proangiogenic and proinflammatory protein expression modulation [9]. On the other hand, there is evidence of its relationship with the activation of brown adipose tissue (BAT) adipogenesis [10], which in turn is associated with metabolic health maintenance. In our study, miR-378 expression was increased in obesity, but exclusively in the metabolically unhealthy phenotype. When dividing the patients into MHO and MUO subgroups, a significant increase in miR378 expression compared to that in the HCG was noted only for patients with MNO. In the MHO group, its expression, on the contrary, positively correlated with the HDL level. These results are consistent with other studies, including the study by Choi et al. [29], where significant positive correlations of miR-378 expression with the anthropometric parameters BMI and WC and the metabolic parameters leptin, insulin, HOMA-IR, TGs, and LDL-C, as well as a negative relationship with HDL were revealed. The data obtained may characterize mir-378 as a biomarker of metabolic disorders in obesity. The differences in the data can be explained by the systemic role of miR378, which consists in the implementation of incoming energy by all available ways—both by its deposition in WAT (white adipose tissue) and by its expenditure in BAT. However, under conditions of a pronounced excess of incoming energy, its physiological role is disrupted, and compensatory hyperexpression of miR378 in white AT and associated metabolic consequences develop. Among the factors contributing to the hyperexpression of miR378, an increase in the level of proinflammatory cytokines can be noted [9]. The fact that miR-378 is significantly elevated only in MUO allows us to consider it as a marker associated with the transformation of MHO into MUO.

These data suggest that it is reasonable to consider miR-142-3p, miR378, and miR-155 not only as promising biomarkers of the metabolically unhealthy phenotype of obesity, but also as targets for intervention in the treatment of metabolic dysregulation associated with obesity.

We found a significant decrease in ADIPOQ gene expression in obese patients compared to healthy controls. This may to some extent explain the transformation of adipose tissue expansion from hyperplastic to hypertrophic. A study by Korac et al. [30] also showed that patients in the MUO group had ADIPOQ expression deficiency in subcutaneous adipose tissue, which was caused by a combination of effects: inflammatory cascades (TNF-α/NF-κB), adipocyte degeneration, hypoxia, and some epigenetic changes. These mechanisms explain the role of adiponectin as a factor with protective properties against insulin resistance and dyslipidemia. We also noted a decrease in ADIPOQ expression in an earlier study [31]. At the same time, the level of ADIPOQ expression in both subcutaneous and visceral ATs was not associated with the level of this adipokine in the circulation. In addition, ADIPOQ expression did not increase with significant weight loss one year after bariatric intervention. This may indicate both complex multilevel mechanisms of the relationship between gene expression in tissues and the level of protein in the circulation and low reversibility of the changes in genetic expression that occur with obesity, i.e., the formation of “cellular memory”.

In general, the data obtained contribute to understanding the chronology and implementation paths of the inflammatory, fibrotic changes in the subcutaneous adipose tissue in obesity and the development of metabolic disorders. Inflammation characterized adipose tissue in obesity even in the absence of metabolic changes, while an increase in hypoxia markers and miR378 was specific for patients with a metabolically unhealthy phenotype, which allows us to consider these markers as signals for transformation into a metabolically unhealthy phenotype. The severity of suppression of antifibrogenic miR-142 was most significant in MUO, which logically indicates the contribution of the loss of this protective mechanism to the progression of adipose tissue fibrosis in MUO, a violation of its depositing capacity, which naturally leads to a switch in fat deposition to visceral deposition and the development of metabolic disorders. The dynamic assessment of these markers in obese patients will reveal the possibility of their use for predicting the risk of obesity progression.

## 4. Materials and Methods

### 4.1. Clinical Characteristics of the Examined Patients

The study included 39 patients with obesity who were diagnosed and treated at the V.A. Almazov National Medical Research Center. All participants provided informed, voluntary consent to take part in the study. The control group consisted of 10 conditionally healthy individuals who were matched in basic demographic parameters (age and gender) with the study participants. These individuals were selected from the staff of the hospital (Table 2). The selection of patients in the obese group was based on the following criteria.

Inclusion criteria: men and women over 18 years old, body mass index (BMI) ≥ 30 kg/m^2^, documented patient’s consent to participate in the study, absence of secondary causes of obesity. Exclusion criteria: significant cardiovascular pathology, i.e., history of myocardial infarction (MI), acute cerebrovascular accident (CVA), angina pectoris, chronic heart failure (CHF) above II functional class (NYHA), high-risk arrhythmias, diabetes presence, chronic kidney disease with glomerular filtration rate (GFR) < 60 mL/min, liver failure, more than 3-fold increase in liver transaminases (ALT, AST), diseases accompanied by changes in thyroid function, as well as the presence of any other serious diseases that could cause changes in the parameters being studied, therapy with immunosuppressants, immunomodulators, or biological drugs received for any reason at the start of the study, indications of alcohol abuse, a history of surgical treatment for obesity.

A comprehensive medical history was collected for each patient, including the duration of obesity, comorbidities, and family history of obesity. Medication use was carefully reviewed to exclude drugs and supplements known to influence brown adipose tissue (BAT) activity, such as beta-blockers, sympathomimetics, mirabegron, adenosine, fibrates, and resveratrol.

Anthropometric measurements were performed, including body weight, body mass index (BMI), waist circumference (WC), hip circumference (HC), and derived ratios (HC/WC and WC/height). Resting heart rate (HR) and blood pressure (BP) were also recorded. Obesity severity was classified according to WHO criteria based on BMI (kg/m^2^) in Grade 1 obesity: BMI 30.00–34.99; Grade 2 obesity: BMI 35.00–39.99; Grade 3 obesity: BMI ≥ 40.00

The division of patients into metabolically healthy obesity and metabolically unhealthy obesity groups was performed based on the Aguilar-Salinas criteria [3], according to which the metabolically healthy obesity group included patients who did not have dyslipidemia (TG level ≤ 150 mg/dL (1.7 mmol/L)-HDL ≥ 40 mg/dL (>1.0 mmol/L for men and >1.2 mmol/L for women)), dysglycemia (<126 mg/dL (7 mmol/L) without antidiabetic drug therapy), and arterial hypertension (<140/90 without antidiabetic drug therapy). As a result, 19 patients were included in the MHO group, and 20 patients in the MUO group.

### 4.2. Laboratory Methods

Metabolic parameters were assessed in all patients included in the study (lipid profile, fasting glucose, and insulin levels with HOMA IR index (insulin resistance index) and HOMA B index (insulin secretion index) calculation, uric acid, CRP). CRP was assessed using an automatic analyzer (Cobas c311, Basel, Switzerland). Serum insulin levels were measured using an automatic analyzer (Cobas e411, Roche, fully automated immunochemical electrochemiluminescence analyzer, Basel, Switzerland) and using the commercial kit “Insulin Elecsys, Cobas e” (Roche, Basel, Switzerland), with a measurement range of 1.39–6945 μIU/mL and a normal value of 17.8–173.0 pmol/L. Conversion factor: pmol/L × 0.144 = μIU/mL.

### 4.3. Instrumental Methods

Also, all subjects included in the study underwent subcutaneous fat fine-needle biopsy by puncturing and aspirating subcutaneous fat from the anterior abdominal wall. The obtained samples were placed in a tank with liquid nitrogen (−196 °C) for transportation and then stored in a refrigerator at −80 °C until analysis. To isolate RNA from the adipose tissue, homogenization of the samples was performed using Tissue Lyser LT and ExtractRNA reagent (Eurogen, Moscow, Russia). For better visualization of low-concentration RNA precipitate in isopropanol, 1 μL of glycogen was added. A NanoDrop spectrophotometer (ThermoFisher, Waltham, MA, USA) was used to measure RNA quality and quantity. The concentration in all samples was 0.5 ng/μL. Analysis of messenger RNA (mRNA) and microRNA (miRNA) levels was performed using quantitative real-time PCR. Reverse transcription and real-time amplification were performed using a Veriti 96-Well Thermal Cyclermodel 9902 (Life Technologies, Carlsbad, CA, USA) and a 7500 Real-Time PCR System (LifeTechnologies, Carlsbad, CA, USA).

Reverse transcription of mRNA was performed using random primers, and reverse transcription of microRNA was performed using a hairpin primer. Hydrolysis probe sets with primers were used to detect the mRNAs of the Adipoq, UCP1, UCP3, Prdm16, Hif1a, and miR155 genes. Detection of the Ccl2 gene mRNA was performed using primers and the qPCR mix-HSSYBR+ROX mastermix reagent (Evrogen, Moscow, Russia). The relative gene expression levels (RQ) in each sample were calculated as RQ = 2^(Cq max − Cq sample).

Total miR142 and miR378 RNA were isolated from adipose tissue biopsies using the ExtractRNA reagent for total RNA isolation from biological samples (Eurogen, cat. No. BC032, Moscow, Russia). Reverse transcription was performed using the TaqMan™ MicroRNA Reverse Transcription Kit (Thermo Fisher Scientific, Waltham, MA, USA) according to the manufacturer’s recommendations. A Veriti 96-well thermal cycler (Applied Biosystems, Foster City, CA, USA) was used to set up the reverse-transcription reaction of miR142 and miR378 microRNAs. Amplification of microRNA by quantitative PCR was performed using TaqMan Universal Master Mix II (Life Technologies, Carlsbad, CA, USA-cat. No. 4440040) and TaqMan probe (Life Technologies, Waltham, MA, USA) according to the manufacturer’s recommendations. To determine the operational range of the detection system for microRNA analysis, a standard curve (a graph of the dependence of the threshold cycles on the logarithm of concentrations in a series of sample dilutions) was constructed. Synthetic oligoribonucleotides (Synthol) with a known concentration identical to that of the analyzed microRNA were used to construct the standard curve. Since reference intervals for these parameters have not been established, a comparison was made with the healthy control group (HCG) (Table 4).

### 4.4. Statistical Analysis

Statistical analysis was performed using STATISTICA 10 (StatSoft Inc., Tulsa, OK, USA) for Windows. The data are presented as mean ± standard deviation. The distribution of the studied variables deviated from normal (a normality test was performed using the Kolmogorov–Smirnov criterion; dispersion was calculated using the Levene’s test). Due to the fact that the distribution in the groups for some parameters was unequal, the use of simple methods of calculation (Student’s t criterion, etc.) was inappropriate. Therefore, we used non-parametric statistical methods to evaluate the studied groups (the Mann–Whitney test was used to compare two independent samples with an interval scale; Wilcoxon’s t criterion was used to evaluate the relationship between a factor and a dependent variable to compare two dependent samples with each other by the level of expression of a feature).

### 4.5. Limitations of the Study

The main limitation of this study is the relatively small number of participants, which may have affected the statistical power of the analysis. In order to increase the reliability of the results and more accurately stratify the patients, further research with a larger sample size and an increased number of comparison groups is needed. However, for the purposes pursued in our work, the number of patients was sufficient, and we also tried to ensure that the samples were representative.

## Figures and Tables

**Figure 1 ijms-26-08212-f001:**
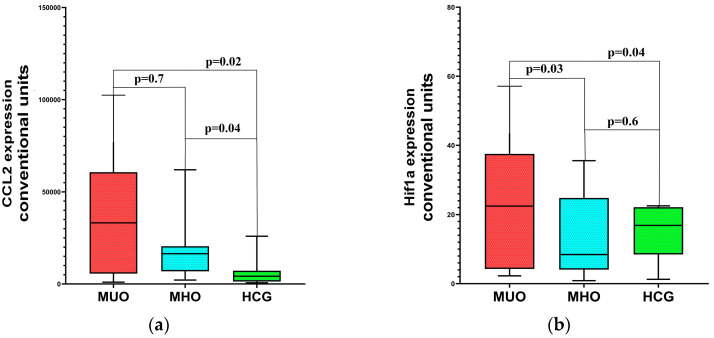
Differences in molecular genetic markers’ expression in MHO and MUO: (**a**)—CCL2 expression, (**b**)—HIF1a expression, (**c**)—miR-378 expression, (**d**)—miR-142 expression, (**e**)—miR-155 expression. *p* at the top—differences between MUO and MHO.

**Table 1 ijms-26-08212-t001:** Aguilar-Salinas criteria of MUO.

Criteria	Aguilar-Salinas
FPG, mg/dL	<126 and no treatment
BP, mmHg	<140/90 and no treatment
TGs, mg/dL	≥150
HDL, mg/dL	≥40
MHO criteria	All of the above
BMI, kg/m^2^	≥30

Note: FPG—fasting plasma glucose; BP—blood pressure; TGs—triglycerides; HDL—high-density lipoprotein; MHO—metabolically healthy obesity; BMI—body mass index.

**Table 2 ijms-26-08212-t002:** Key characteristics of obese and healthy control groups.

Groups	Age, Years	M/F (%)	BMI (kg/m^2^)	FGL, mmol/L	SBP, mmHg	DBP, mmHg	HDL-C, mmol/L	TGs, mmol/L
Obesity, *n* = 39	42.0 (30.0–53.0)	23.1/76.9	35.71 (32.2–40.3)	5.47 (5.16–5.67)	124.8 (113.4–133.2)	71.3 (70.0–82.4)	1.42 (1.0–1.68)	1.37 (1.0–1.76)
Healthy control, *n* = 10	38.9 (29–49)	18.2/81.8	23.3 (22.1–24.9)	4.71 (4.4–5.4)	116.8 (111.3–128.8)	77.3 (71.1–85.4)	1.4 (1.3–1.51)	1.29 (1.01–1.68)
*p*	>0.05	>0.05	≤0.001	≤0.05	>0.05	>0.05	>0.05	>0.05

Note: M/F—male/female ratio; BMI—body mass index; FGL—fasting glucose level; SBP—systolic blood pressure; DBP—diastolic blood pressure; HDL-C—high-density lipoprotein cholesterol; TGs—triglycerides.

**Table 3 ijms-26-08212-t003:** Characteristics of clinical and metabolic parameters in obesity patients and healthy controls included in the study.

Assessed Parameters		Obese Patients *n* = 39	Reference or Target Intervals, Where Applicable
Age, years		42.0 (30.0–53.0)	NA
Body weight, kg		102.0 (91.0–116.0)	NA
BMI, kg/m^2^		35.71 (32.2–40.3)	<25
WC, cm	Male and female	108.0 (100.0–118.0)	
	Male	118.0 (117–120.0)	<90
	Female	107.0 (96.0–114.0)	<84
HC, cm	Male and female	119.0 (112.0–126.0)	
	Male	114.0 (108.0–121.0)	NA
	Female	121.0 (112.0–128.0)	NA
Fasting glucose level, mmol/L		5.47 (5.16–5.67)	3.3–6.1
Fasting insulin level, pmol/L		141.05 (80.9–182.8)	17.8–173.0
HOMA-IR		4.72 (2.69–7.85)	<2.77
HOMA-B		202.0 (132.9–288.9)	>100
TC, mmol/L		5.38 (4.3–5.78)	<4.5
HDL-C, mmol/L	Male and female	1.42 (1.0–1.68)	
	Male	1.12 (0.99–1.43)	>1.0
	Female	1.47 (1.08–1.68)	>1.2
LDL-C, mmol/L		2.89 (2.23–3.61)	<1.8
TGs, mmol/L		1.37 (1.0–1.76)	<1.7
SBP, mmHg		124.8 (113.4–133.2)	<140
DBP, mmHg		71.3 (70.0–82.4)	<85

Note: BMI—body mass index; WC—waist circumference; HC—hip circumference; HOMA-IR—homeostasis model assessment of insulin resistance; HOMA-B—homeostasis model assessment—pancreatic B-cell function; TC—total cholesterol; HDL-C—high-density lipoprotein cholesterol; LDL-C—low-density lipoprotein cholesterol; TGs—triglycerides; SBP—systolic blood pressure; DBP—diastolic blood pressure.

**Table 4 ijms-26-08212-t004:** Genetic markers’ expression in SAT in patients with obesity in comparison with the healthy control group.

Indicant	Obesity Before Treatment, *n* = 39	Healthy Control, *n* = 10	*p*
ADIPOQ	3.01 (2.18–3.89)	6.305 (2.3–7.68)	0.039
HIF1a	12.95 (4.39–30.2)	20.95 (11.62–22.550)	0.63
CCL2	13,829 (6861–50,704)	4200 (1397–5302)	0.004
miR-155	12,960 (6385–19,960)	6304 (6039–8623)	0.017
miR-142	3805 (675–16,300)	21,400 (11,035–39,100)	0.008
miR-378	3360 (1650–5600)	1820 (1131–2620)	0.04

Note: ADIPOQ—adiponectin, C1Q, and collagen domain-containing; HIF1a—hypoxia-inducible-factor 1a; miR-155—microRNA-155; miR-142—microRNA-142; miR-378—microRNA-378; CCL2—C-C motif chemokine ligand 2.

**Table 5 ijms-26-08212-t005:** Characteristics of patients in MHO and MUO groups (calculated according to Aguilar-Salinas).

Parameter	MHO (*n* = 20)	M3O (*n* = 19)	*p*
Age, years	36 (25–47)	39 (30.5–49)	0.1
Duration of obesity, years	9.32 (6.3–12.8)	8,97 (7.7–11.6)	0.5
% male	20.6	21.4	>0.05
BMI, kg/m^2^	34.8 (32.8–39.8)	35.9 (31.2–41.6)	0.2
Glucose, mmol/L	5.6 (5.4–6.0)	5.36 (5.12–5.6)	0.05
Insulin, pmol/L	160.0(122.1–254.4)	102.3 (63.3–155.3)	0.01
HOMA-IR	5.7 (5.16–5.97)	3.57 (2.2–5.4)	0.02
TC, mmol/L	5.65 (3.96–6.0)	5.26 (4.42–5.78)	0.3
HDL-C, mmol/L	1.0 (0.91–1.49)	1.53 (1.41–1.8)	0.001
TGs, mmol/L	1.8 (1.2–2.34)	1.15 (0.85–1.38)	0.001
SBP, mmHg	128.3 (112–136)	126.4 (109–132.1)	0.6
DBP, mmHg	73 (70–80)	73.3 (70–81.4)	0.1
CRP, mg/L	3.61 (2.0–10.5)	2.49 (0.72–6.1)	0.04

Note: BMI—body mass index; HOMA-IR—homeostasis model assessment of insulin resistance; TC—total cholesterol; HDL-C—high-density lipoprotein cholesterol; TGs—triglycerides; SBP—systolic blood pressure; DBP—diastolic blood pressure; CRP—C-reactive protein.

**Table 6 ijms-26-08212-t006:** Correlation of CCL2 expression with metabolic parameters.

Parameters	CCL2 in the MUO Group		CCL2 in the MHO Group	
	r	*p*	r	*p*
Glucose	0.63	0.001	−0.48	0.26
TC	0.45	0.03	−0.02	0.9
TGs	0.52	0.01	0.3	0.5

Note: TC—total cholesterol, TGs—triglycerides, MUO—metabolically unhealthy obesity, MHO—metabolically healthy obesity.

## Data Availability

The original data presented in the study are openly available in [17].

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
