# Peer review of "Comparative Analysis of Gene and MicroRNA Expression in Subcutaneous Adipose Tissue in Metabolically Healthy and Unhealthy Obesity"

_ijms, 2025, doi:10.3390/ijms26178212_

Round 1

Reviewer 1 Report

Comments and Suggestions for Authors

This paper tries to reveal key pathogenetic differences between these phenotypes (Metabolically healthy and unhealthy obesity, MHO and MUO) through studying gene and microRNA expression in subcutaneous adipose tissue (SAT). This research is meaningful. However, there are some suggestions for the study:

  1. Molecular genetic Background in the title should be more specific, and include studying gene and microRNA expression in subcutaneous adipose tissue (SAT). like molecular genetic characteristics of SAT.
  2. In addition, how to determine MHO and MUO should be supported by some properties, like insulin resistance or hyperlipemia, because healthy or not is general.
  3. In the abstract, the authors aim to reveal exhibit distinct molecular genetic mechanisms underlying metabolic disorders by comparing the metabolically healthy (MHO) and unhealthy obesity (MUO) groups. Based on the hypothesis that blood supply deficit and hypoxia increase-caused inflammation and fibrosis in SAT limits its plasticity further expansion, the expression of genes (ADIPOQ, HIF1A, CCL2) and microRNAs (miR-142-3p, miR-155, miR-378) in SAT were detected and analyzed with biochemical properties. However, though most of these expression changes existed between the obese group and the control group, no significant change was found between the MHO group and MUO group. Obviously, the results failed to support the idea that blood supply deficit and hypoxia in SAT resulted in or correlated to the different metabolism between the MHO and MUO groups. More properties from other tissue might be need to be screened to support the hypothesis. Therefore, the manuscript must be re-written, the current conclusion is not successful for the title and the purpose of the research.
  4. Line 38-42, the sentence is too long.
  5. Where are Participants from should be introduced.
  6. The gender information of healthy control should also be needed.
  7. In table 1, age years 0 (30.0; 53.0), ; should be replaced by -, so does BMI and other properties.
  8. Line 166, at the end, a full stop is missed.
  9. In figure 1, the error bar is extremely huge? The data is not reliable any more.

Author Response

  1. Molecular genetic Background in the title should be more specific, and include studying gene and microRNA expression in subcutaneous adipose tissue (SAT). like molecular genetic characteristics of SAT.

Response: Dear reviewer, thank you for your careful consideration of our article and valuable comments. We have taken into account your suggestion regarding the title change and have made adjustments. The new title of the article is: "Comparative analysis of gene and microRNA expression in subcutaneous adipose tissue in metabolically healthy and unhealthy obesity"

  1. In addition, how to determine MHO and MUO should be supported by some properties, like insulin resistance or hyperlipemia, because healthy or not is general.

Response: We would like to express great gratitude for your comment. We would also like to apologize for any confusion that may have been caused by the absence of a clear definition of patient criteria. To address this, we have added criteria to differentiate between patients with MHO and MUO, and we have attached a link in the text of the article that explains these changes.

Line 50-55.

  1. In the abstract, the authors aim to reveal exhibit distinct molecular genetic mechanisms underlying metabolic disorders by comparing the metabolically healthy (MHO) and unhealthy obesity (MUO) groups. Based on the hypothesis that blood supply deficit and hypoxia increase-caused inflammation and fibrosis in SAT limits its plasticity further expansion, the expression of genes (ADIPOQ, HIF1A, CCL2) and microRNAs (miR-142-3p, miR-155, miR-378) in SAT were detected and analyzed with biochemical properties. However, though most of these expression changes existed between the obese group and the control group, no significant change was found between the MHO group and MUO group. Obviously, the results failed to support the idea that blood supply deficit and hypoxia in SAT resulted in or correlated to the different metabolism between the MHO and MUO groups. More properties from other tissue might be need to be screened to support the hypothesis. Therefore, the manuscript must be re-written, the current conclusion is not successful for the title and the purpose of the research.

Response: We agree that increasing the number of groups would increase the significance of the differences. We have added this limitation to the relevant section of our article, as follows:

"One limitation of our study is the relatively small number of groups analyzed, which may have affected the statistical power of our analysis. Expanding the sample size by including additional observation groups would be a promising direction for future research." There is no doubt that increasing the number of patients in the groups under study may lead to more accurate results.

However, for the purposes we pursued in our work, the number of patients is sufficient, and we also tried to ensure that the samples were representative.

Line 416-421.

We found significant differences in the expression of HIF-a and miR-378 markers in patients with MUO, compared to the healthy control group. Therefore, we have allowed ourselves to express an opinion on the importance of these differences in expression in our conclusion.

  1. Line 38-42, the sentence is too long.

Response: Thank you for your feedback. We have made some changes to the sentence to improve clarity and have reduced the number of words.

Line 40-44.

  1. Where are Participants from should be introduced.

Response: Thank you very much for your corrections. We appreciate your comment and we understand that it is important for our readers to know where the study participants came from. We have now added this information to our Materials and Methods section: «The study included 39 patients with obesity who underwent examination and treatment at the V.A. Almazov National Medical Research Center. All participants provided informed, voluntary consent to take part in the study. The control group consisted of 10 conditionally healthy individuals who were matched in basic demographic parameters (age and gender) with the study participants. These individuals were selected from the staff of the hospital».

Line 319-324.

  1. The gender information of healthy control should also be needed.

Response: Thank you for your feedback. The information regarding the gender ratio of obese patients compared to healthy controls can be found in Table 2.

Line 92.

  1. In table 1, age years 0 (30.0; 53.0), ; should be replaced by -, so does BMI and other properties.

Response: Thank you, we’ve corrected it. Now you may see it in table 2.

Line 92.

  1. Line 166, at the end, a full stop is missed.

Response: Thank you for your thoughtful review, we have made the necessary edits.

Line 178.

  1. In figure 1, the error bar is extremely huge? The data is not reliable any more.

Response: Thank you very much for your feedback. We have added confidence intervals for better clarity and provided the requested drawings.

Line 151-157.

Reviewer 2 Report

Comments and Suggestions for Authors

Mostly clinical study, describing correlations between healthy and pathologic obesity in relatively small groups of patients and healthy controls. Evaluation of different diagnostic parameters and data on patients are sufficient and sound. Biopsy samples of subcutaneous fat were used for RNA isolation and real-time PCR analysis of particular mRNAs encoding adiponectin, HIF-1a, CCL2, and several microRNAs implicated in obesity and control of adipocyte growth and development. Selection of these markers were based on literature data and seem logical. The authors did correlation analysis and found that inflammation-related chemokine CCL2 gene expression directly correlate with pathologic infiltration of macrophages into fat tissue. Despite novelty and interesting observations, the manuscript has several issues that should be addressed prior to acceptance for publication.

Major points:

1) It seems that numbers of individuals in the groups could be increased, and this should increase the soundness of statistics provided by the authors;

2) The figure showing histoplots of relative gene expression is fine, but it would be more meaningful to re-plot the data in the format of box plots with whiskers. That would give readers more info concerning variations between patients;

3) Is it possible to make tissue extracts from biopsy samples of fat to measure actual levels, for example, of CCL2 and adiponectin. It would greatly complement RNA analysis data. I would also suggest to perform ELISA analysis of blood levels of adiponectin and CCL2, and possible, TNFa. That would show not only local status of fat inflammation, but also systemic global changes;

4) Not sure that it would be easy, but histologic analysis of selected samples of fat biopsy from all groups used in the study would signicantly increase the value of the data by bringing together molecular data and analysis of different cell types inside the fat tissue;

5) Scientific English of the manuscript could be improved, there are multiple grammar errors that should be corrected by professional editor or native English speaker.

Minor point: it looks like on top of table 4 English abbreviation MUO was mistakingly substituted by Cyrillic abbleviation.

Verdict: major revision with point-by-point response to issues.  

Comments on the Quality of English Language

Englished should be improved by professional editing of the text. 

Author Response

Major points:

  • It seems that numbers of individuals in the groups could be increased, and this should increase the soundness of statistics provided by the authors;

Response: There is no doubt that increasing the number of patients in the groups under study may lead to more accurate results. However, for the purposes we pursued in our work, the number of patients is sufficient, and we also tried to ensure that the samples were representative.

  • The figure showing histoplots of relative gene expression is fine, but it would be more meaningful to re-plot the data in the format of box plots with whiskers. That would give readers more info concerning variations between patients;

Response: Thank you very much for your feedback. We have provided the requested drawings.

Line 151-157.

  • Is it possible to make tissue extracts from biopsy samples of fat to measure actual levels, for example, of CCL2 and adiponectin. It would greatly complement RNA analysis data. I would also suggest to perform ELISA analysis of blood levels of adiponectin and CCL2, and possible, TNFa. That would show not only local status of fat inflammation, but also systemic global changes;

Response: We fully support your proposal to conduct additional experiments by forming tissue extracts from biopsy fat samples in order to measure the actual levels of, for example, CCL2, adiponectin.

Unfortunately, as part of the current study, we were unable to perform these analyses due to the limited volume of biomaterial and time frame. However, we consider your recommendations extremely valuable and plan to include them in further research. This will not only confirm the RNA analysis data, but also provide a more complete picture of metabolic and inflammatory changes in obesity. Thank you again for your attention to our work and valuable recommendations. We will definitely take them into account in our future projects.

However, data on the comparison of changes in the levels of microRNAs in plasma and adipose tissue was presented in our previous publication (Matveev G.A., Khromova N.V., Zasypkin G.G., Kononova Ya.A., Vasilyeva E.Yu., Babenko A.Yu., Shlyakhto E.V. Tissue and circulating microRNAs 378 and 142 as biomarkers of obesity and reactions to its treatment. Publication date: August 30, 2023;24(17):13426. doi: 10.3390/ijms241713426. PMID: 37686231; PMCID: PMC10487855).

  • Not sure that it would be easy, but histologic analysis of selected samples of fat biopsy from all groups used in the study would signicantly increase the value of the data by bringing together molecular data and analysis of different cell types inside the fat tissue;

Response: Thank you for your careful consideration of our work and valuable comments, which undoubtedly contribute to increasing its scientific value.

Regarding your proposal to conduct a histological examination of biopsy samples of adipose tissue, we would like to clarify that the biopsy technique used in our study (a biopsy of the pancreas will be performed by puncturing subcutaneous fat in the anterior abdominal wall using injection needles (18G (1.20*40 mm)) using the "free hand" method) does not provide for the collection of material in a volume sufficient for qualitative histological analysis. The main purpose of the biopsy in our protocol was to obtain material for molecular genetic studies (isolation of microRNAs, genes), which determined the minimally invasive nature of the procedure.

We are fully aware of the importance of histological assessment, which could supplement our data with information about morphological changes in adipose tissue. This is indeed of considerable scientific interest. In the future, we plan to modify the biomaterial sampling protocol to include histological analysis in future studies.

  • Scientific English of the manuscript could be improved, there are multiple grammar errors that should be corrected by professional editor or native English speaker.

Response: We sincerely appreciate your valuable feedback regarding the language quality of our manuscript. Following your suggestion, we have carefully reviewed the text and implemented comprehensive grammatical and syntactical corrections to improve the clarity and accuracy of the Scientific English.

We hope that the improvements made address your concerns and enhance the overall readability of the manuscript.

Minor point: it looks like on top of table 4 English abbreviation MUO was mistakingly substituted by Cyrillic abbleviation.

Response: Thank you, we’ve corrected it. Now you may see it in table 5

Line 142.

Round 2

Reviewer 1 Report

Comments and Suggestions for Authors

The manuscript is geartly improved.

However, the mRNA expression result in figure 1 was still confusing due to the high SD value. We usually don't have such a huge SD in mRNA expression, some of them were 2 times larger than the average value. I recomend to detect these data again.

In addition, we use a normalized data with the control average value of 1.0.

Author Response

Dear Reviewer, We thank you for your careful consideration of our work and valuable comments. We have carefully studied your recommendations and made appropriate adjustments, in particular, by performing standardization in accordance with your instructions.

Line 152-153